# Exogenous Ethylene Alleviates the Inhibition of *Sorbus pohuashanensis* Embryo Germination in a Saline-Alkali Environment (NaHCO_3_)

**DOI:** 10.3390/ijms24044244

**Published:** 2023-02-20

**Authors:** Yutong Wang, Caihong Zhao, Xiaodong Wang, Hailong Shen, Ling Yang

**Affiliations:** 1State Key Laboratory of Tree Genetics and Breeding, School of Forestry, Northeast Forestry University, Harbin 150040, China; 2State Forestry and Grassland Administration Engineering Technology Research Center of Korean Pine, Harbin 150040, China; 3State Forestry and Grassland Administration Engineering Technology Research Center of Native Tree Species in Northeast China, Harbin 150040, China

**Keywords:** ethylene, embryo germination, saline-alkaline stress, *Sorbus pohuashanensis*, physiological characteristics, germination parameters, phytohormones

## Abstract

Saline-alkali stress is a major environmental stress affecting the growth and development of plants such as *Sorbus pohuashanensis.* Although ethylene plays a crucial role in plant response to saline-alkaline stress, its mechanism remains elusive. The mechanism of action of ethylene (ETH) may be related to the accumulation of hormones, reactive oxygen species (ROS), and reactive nitrogen species (RNS). Ethephon is the exogenous ethylene donor. Therefore, for the present study we initially used different concentrations of ethephon (ETH) to treat *S. pohuashanensis* embryos and identified the best treatment concentration and method to promote the release of dormancy and the germination of *S. pohuashanensis* embryos. We then analyzed the physiological indexes, including endogenous hormones, ROS, antioxidant components, and reactive nitrogen, in embryos and seedlings to elucidate the mechanism via which ETH manages stress. The analysis showed that 45 mg/L was the best concentration of ETH to relieve the embryo dormancy. ETH at this concentration improved the germination of *S. pohuashanensis* by 183.21% under saline-alkaline stress; it also improved the germination index and germination potential of the embryos. Further analysis revealed that ETH treatment increased the levels of 1-aminocyclopropane-1-carboxylic acid (ACC), gibberellin (GA), soluble protein, nitric oxide (NO), and glutathione (GSH); increased the activities of superoxide dismutase (SOD), peroxidase (POD), catalase (CAT), nitrate reductase (NR), and nitric oxide synthase (NOS); and decreased the levels of abscisic acid (ABA), hydrogen peroxide (H_2_O_2_), superoxide anion, and malondialdehyde (MDA) of *S. pohuashanensis* under saline-alkali stress. These results indicate that ETH mitigates the inhibitory effects of saline-alkali stress and provides a theoretical basis by which to establish precise control techniques for the release of seed dormancy of tree species.

## 1. Introduction

Saline-alkali stress, one of the most common abiotic stresses, affects various plant growth and development processes, from seed germination to flowering and fruiting. This stress lowers and delays germination, causes high seedling mortality, and decreases chlorophyll content and photosynthetic performance [1]. Under saline-alkaline stress, seeds absorb less water, limiting proteolytic enzyme activity, restricting respiration, and reducing germination [2,3]. In plants, ethylene is an important hormone and signal substance that participates in many physiological and biochemical processes related to growth and development. It is also involved in plant response to biotic and abiotic stresses [4].

Saline-alkali stress reduces plant biomass by affecting plant photosynthesis, enzyme activity, and stomatal opening [5]. Under sodium bicarbonate (NaHCO_3_) stress, wheat has been found to accumulate malic acid by upregulating malate dehydrogenase (MDH) and downregulating malate enzyme and tricarboxylic acid (TCA) as part of a cycle involved in coping with alkaline stress [6]. Similarly, cotton regulated the secondary metabolite pathway to maintain ion homeostasis and promote tolerance to NaHCO_3_ stress [7]. Studies have demonstrated that exogenous ethylene effectively mitigates the inhibitory effects of salt stress on seed germination; however, the underlying regulatory mechanism remains unclear [8].

*Sorbus pohuashanensis* is a deciduous tree belonging to Maloideae, a member of the Rosaceae subfamily. It is a tree species with high ornamental, medicinal, and edible values [9]. However, the wild *S. pohuashanensis* population is less dense and sensitive to interspecific competition, which results in poor natural regeneration [10,11,12]. Furthermore, the overexploitation of *Sorbus,* with immense ornamental value, has caused severe damage to the wild resources in the forests of northeastern China; consequently, the population has been endangered in certain areas.

Previous studies have demonstrated that ethephon (ETH), the ethylene donor, significantly improved the germination percentage, index, and potential of *S. pohuashanensis* embryos [13]. Additionally, nitric oxide (NO) has been shown to alleviate *S. pohuashanensis* embryo dormancy, which was closely related to ethylene biosynthesis and abscisic acid (ABA) catabolism [14]. Spraying exogenous ethylene mitigated the inhibitory effect of salt stress on seed germination effectively [15]. In Arabidopsis, excessive endogenous ethylene synthesis and ACC (1-amino-cyclopropane-1-carboxylic acid) treatment reduced salt stress-induced inhibition of seed germination. Typically, ethylene synthesis could be regulated by targeting ACC synthase (ACS) and ACC oxidase (ACO), which are closely related to these hormones. They together regulate the signal transduction and physiological processes of ethylene. Numerous studies have also reported that exogenous ethylene help plants cope with salt stress by regulating the synthesis and removal of reactive oxygen species (ROS) [8]. Though ROS accumulation breaks dormancy and seed germination in certain species [16,17], conditions such as salinity and alkalinity lead to a sharp increase in the production rate of ROS, resulting in oxidative stress [18]. In, hydrogen peroxide (H_2_O_2_), a representative ROS was accumulated more in germinating seeds under saline-alkaline conditions than in non-saline-alkaline states, significantly delaying germination [19].

Thus the present study, in which we initially treated *S. pohuashanensis* embryos with ETH at different concentrations to find the optimal concentration and the method to promote seed dormancy release and germination. We further assessed the effect of exogenous ethylene (ETH) on the germination of *S. pohuashanensis* embryos under saline-alkali stress by analyzing the physiological indexes, including endogenous hormones, ROS, and active nitrogen, in the embryos and seedlings. The results of these assays will help elucidate the mechanism via which ethylene alleviates the dormancy of *S. pohuashanensis* embryos under saline-alkali stress. The study will also provide novel insights into the role of exogenous ethylene in promoting the germination of *S. pohuashanensis* embryos.

## 2. Results

### 2.1. Effects of ETH and NaHCO_3_ on Germination of S. pohuashanensis Embryos

The present study initially examined the effect of different concentrations of ethephon, the ethylene donor, on the germination of *S. pohuashanensis* embryos. The exogenous addition of ETH significantly increased the germination percentage of *S. pohuashanensis* embryos (Figure 1). Treatment with different concentrations of ETH (30, 45, and 60 mg/L) significantly improved the germination percentages of *S. pohuashanensis* (Figure 1). Among the various concentrations, 45 mg/L of ETH resulted in the highest germination of *S. pohuashanensis* embryos (60%), which was significantly higher than that under no ETH treatment (Figure 1).

Germination of *S. pohuashanensis* embryos was severely impaired under salt stress. Here, *S. pohuashanensis* seeds showed only a 6.67% germination percentage when treated with NaHCO_3_. Meanwhile, the percentage of germination in control and ETH treatments was 84.6% and 88.9%, respectively, which were significantly higher than the NaHCO_3_ treatment (*p* ≤ 0.05, Table 1). Further, to analyze the role of ETH in regulating the inhibitory effects of the saline-alkali environment, we treated the seeds with a mixture of ETH and NaHCO_3_. The results show that the embryo germination percentage increased by a generally lower amount than the ETH and NaHCO_3_ treatment when compared with NaHCO_3_ alone. Similar results were also observed for germination index and germination potential (Table 1). These results indicate that ETH reduced the inhibitory effects of saline-alkali stress on the germination of *S. pohuashanensis* embryos (Figure 2).

### 2.2. Changes in Hormone Content of Embryos during Germination in S. pohuashanensis

As the exogenous addition of ETH and NaHCO_3_ showed a difference in the germination percentage of *S. pohuashanensis* embryos, we further analyzed the hormone content of the seeds under various treatments during germination, daily, for eight days (Figure 3). The ACC content of embryos increased from day 1 to 8 under all treatments. Under the ETH treatment, the ACC content first increased, followed by a slight decrease, and then an increase (Figure 3A). On day 8, the content of ACC in seeds treated with ETH was significantly higher than that under the other treatments (Figure 3A). After the addition of ETH to NaHCO_3_, the release of ACC significantly increased. These results indicate that the exogenous addition of ETH promoted the release of ACC in embryos under NaHCO_3_ stress. Similarly, the ABA content also increased with time during the release of dormancy and germination of seeds under all treatments (Figure 3B). The ABA content of the seeds was in the following order: NaHCO_3_ > NaHCO_3_ + ETH > CK > ETH (Figure 3B; *p*). Furthermore, ETH treatment significantly increased the GA content in the embryos (Figure 3C). In the early stage of germination, the differences in GA content of seeds among the treatments were insignificant. By day 8, ETH alone resulted in high GA content, while NaHCO_3_ resulted in a low GA content; however, ETH added to NaHCO_3_ resulted in high GA content (Figure 3C).

### 2.3. Changes in ROS during Germination of S. pohuashanensis Embryos

Further, to study the effects of ETH on ROS during embryo germination of *S. pohuashanensis* under NaHCO_3_ stress, we evaluated the changes in peroxide (H_2_O_2_), superoxide anion, and malondialdehyde (MDA) content (Figure 4). During the late germination stage, control and ETH-treated seeds had almost similar H_2_O_2_ levels; NaHCO_3_ treatment significantly triggered the accumulation of H_2_O_2_, while the presence of ETH under NaHCO_3_ stress reduced the accumulation of H_2_O_2_ (Figure 4A). This proved that the combination of both treatments showed an antagonistic effect, which significantly reduced the H_2_O_2_ content in the embryos.

Furthermore, all treatments reduced the superoxide anion content during the middle and late stages of embryo germination (Figure 4B). The superoxide anion content of *S. pohuashanensis* embryos treated with NaHCO_3_ was significantly higher compared with those under ETH alone and in combination with NaHCO_3_. This observation indicates that adding ETH did not improve the superoxide anion content of *S. pohuashanensis* embryos subjected to NaHCO_3_. In addition, treatment with ETH alone resulted in lower MDA levels than the control, while NaHCO_3_ treatment resulted in high MDA levels (Figure 4C). However, ETH added to NaHCO_3_ reduced the MDA content of the embryos by 48.68%.

### 2.4. Changes in Antioxidant Enzyme Activities during Germination of S. pohuashanensis Embryos

We further determined the levels of SOD, CAT, POD, soluble protein, and GSH to assess the effect of exogenous ETH and NaHCO_3_ on the antioxidant system during the germination of *S. pohuashanensis* embryos. Seedlings under NaHCO_3_ stress showed lower SOD, CAT, and POD activities, soluble proteins, and GSH content than the other treatments (Figure 5).

The SOD activity of *S. pohuashanensis* embryos gradually increased with the release of embryo dormancy and germination. The SOD levels in embryos treated with ETH were consistently higher than those under control (CK), while those under NaHCO_3_ + ETH treatment were higher than those under NaHCO_3_ alone (Figure 5A). The POD activity in the embryos treated with NaHCO_3_ increased initially and then decreased, while that in the embryos under other treatments showed a gradual increase (Figure 5B). The addition of ETH increased POD activity by 56.44% compared with the control (Figure 5B). Furthermore, the addition of ETH to NaHCO_3_ resulted in a 50.25% increase in embryonic POD activity. ETH-treated *S. pohuashanensis* embryos showed a gradual increase in CAT activity (Figure 5C), while the NaHCO_3_ and NaHCO_3_ + ETH treatments showed an increase initially and then a decrease. The embryos under NaHCO_3_ stress had CAT levels lower than the control; however, the addition of ETH to NaHCO_3_ resulted in a significant increase in the CAT activity compared with those under NaHCO_3_ treatment alone (Figure 5C).

The soluble protein content of the *S. pohuashanensis* embryos treated with ETH increased. The ETH treatment increased the soluble protein content of embryos compared with the control, while NaHCO_3_ decreased. Furthermore, the addition of ETH to NaHCO_3_ increased the soluble protein content of embryos by 49.17% (Figure 5D). At the start of germination, the soluble protein content of treatment with ETH showed 1.79 μmol/g FW, which was significantly reduced to 0.77 μmol/g FW under NaHCO_3_ treatment. Further, there was an 80.52% increase in the glutathione (GSH) content of the embryos when ETH was added to NaHCO_3_ (Figure 5E). During the late germination stage (eighth day), the GSH content of *S. pohuashanensis* embryos treated with NaHCO_3_ was the lowest (0.46 μmol/g FW).

### 2.5. Changes in Active Nitrogen Content during Germination of S. pohuashanensis Embryos

The NO content in the embryos of *S. pohuashanensis* showed an overall increase towards the late germination stage. NO content in the ETH-treated embryos was the highest, while that under NaHCO_3_ was the lowest (Figure 6A). After the addition of ETH to NaHCO_3_, the NO content of the embryos gradually increased by 23.84% (Figure 6A).

We further analyzed the activity of NR and NOS associated with the conversion of nitrate–nitrogen to ammonia–nitrogen in plants. The NR activity of *S. pohuashanensis* embryos after NaHCO_3_ treatment and the NaHCO_3_ and ETH combined treatment gradually increased with time (Figure 6B). The NR activity of ETH-treated embryos was significantly higher than CK. The addition of ETH to NaHCO_3_ resulted in a 20.29% increase in embryonic NR activity (Figure 6B). In addition, the ETH-treated embryos showed increased NOS activity (Figure 6C). The addition of ETH to NaHCO_3_ resulted in a 15.60% increase in embryonic NOS activity (Figure 6C).

## 3. Discussion

Excessive salinity inhibits plant growth, leading to diminished yield and even death [20]. Seed germination percentage is an important index for identifying the saline-alkaline tolerance of plant species [21] and ETH treatment showed a better effect on seed germination under salt stress [22]. Similarly, in this study, ETH significantly promoted the germination of *S. pohuashanensis* embryos, which NaHCO_3_ inhibited. These observations indicate that ETH significantly alleviated the inhibitory effects of NaHCO_3_ on *S. pohuashanensis* embryos. However, ETH promoted embryo germination at 45 mg/L but not at 30 mg/mL and 60 mg/mL concentrations. These observations suggest 45 mg/mL to be the optimal concentration for promoting germination in *S. pohuashanensis*. NO has also demonstrated a similar concentration-dependent effect on embryo germination in *S. pohuashanensis* [23]. Low concentrations of NO promoted *S. pohuashanensis* embryo germination, while high concentrations inhibited it [23].

In plants, multiple hormones regulate the process of seed germination. Among these phytohormones, ETH regulates salt tolerance by inducing several protective mechanisms [24,25] or enhancing antioxidant systems by increasing nutrient accumulation [26]. Typically, ACC, GA, and ABA are the hormones involved in germination, and the addition of exogenous ETH dramatically increases the release of endogenous ETH [15]. ACC may function as a novel signaling molecule [27]. In this study, ACC was produced continuously with progress in germination under ETH treatment, while the lowest amount of ACC was produced in NaHCO_3_-treated *S. pohuashanensis* embryos. This observation suggests that salt stress could enhance ETH production by increasing the enzyme activity of the ETH biosynthetic pathway [28]. In plants, ABA gets significantly upregulated under salt stress and positively regulates plant responses to stress [29]. Our study found that the endogenous ABA content was lower with the addition of exogenous ETH, while it was always higher under salt stress. This observation indicates the positive role of ABA in regulating the response to salt stress. With the effect of ETH on ABA being opposite, a possible mutual inhibition between ETH and ABA was speculated. Thus, ETH slowed down the embryo’s response to salt stress by reducing the accumulation of ABA, thereby promoting the germination of embryos. In plants, GA is an endogenous signaling molecule breaking seed dormancy. Many studies have demonstrated the antagonistic effect of ABA on GA, thereby inhibiting seed germination [29]. Studies have showed that GA_3_ content decreased with salt stress [30]. In this study, the GA content of *S. pohuashanensis* embryos treated with NaHCO_3_ was significantly lower than those under ETH and ETH + NaHCO_3_ treatments, consistent with previous findings. Meanwhile, ETH alleviated the effects of salt stress on seed germination against seedling growth.

Salt stress severely affects the balance of free radicals in plant cells, leading to the accumulation of ROS, peroxidation of cell membranes, and damage to cell structure [31]. Germination of seeds cannot be uncoupled from the accumulation of ROS either under normal or stress conditions. However, H_2_O_2_ accumulates in seeds under salt stress and exacerbates membrane injury, leading to poor seed germination and even death [32]. In this study, NaHCO_3_ treatment caused significant H_2_O_2_ accumulation, whereas treatment with ETH alleviated H_2_O_2_ accumulation in *S. pohuashanensis* embryos under salt stress.

Antioxidant enzymes and antioxidants, the key factors in ROS detoxification, maintain cellular redox homeostasis within their physiological limits [33]. Under saline-alkaline stress, CAT and SOD play essential roles in defense against oxidative damage and regulate the mutual coordination of antioxidant enzymes to eliminate intracellular ROS. In this study, ETH treatment increased the content or activities of various antioxidant enzymes in the embryos of *S. pohuashanensis* but reduced the accumulation of MDA and H_2_O_2_. They also increased the activity of POD, consistent with earlier findings [15]. A decrease in oxidative stress observed in salt-stressed plants after ETH treatment might be due to increased antioxidant enzyme activities, proline and glutathione content, and redox status. However, Ozt€urk and Demir have reported that ETH decreases POD activity under salt stress [34]. GSH is an important antioxidant that contributes to abiotic stress tolerance; it maintains an intracellular reduced redox environment by metabolizing various ROS and their reaction products. Excess ETH produced by ETH under salt stress was minimized in the optimal range, which favorably regulated GSH production by controlling the enzymatic activity of the ascorbate glutathione cycle [35]. Thus, the study showed that saline-alkaline stress adversely affected the activities of antioxidant enzymes in seedlings; their activities decreased with the severity of stress [31]. However, under saline-alkaline stress, ETH alleviated the germination inhibitory effects by increasing the antioxidant enzyme levels and regulating ROS production.

In addition, saline-alkaline conditions decreased the amount of NO in *S. pohuashanensis* embryos. Meanwhile, the NO amount increased by 23.84% under NaHCO_3_ + ETH treatment compared with NaHCO_3_ treatment. This observation suggests that the germination of *S. pohuashanensis* seeds depends on the action of classical phytohormones, such as GA, and the regulation of signaling molecules, such as NO [36]. In Arabidopsis and apple embryos, NO was found to break seed dormancy and stimulate germination by inducing ETH biosynthesis [16,37,38,39]. Similar results have been demonstrated in the embryos of *S. pohuashanensis* [14]. NR is one of the sources of NO signaling molecules in plants, and enhanced NR activity suggests that ethylene glycol promotion of rowan embryo germination may be closely related to nitric oxide signaling because NR is a source of intracellular nitric oxide signaling molecule production [14]. In *S. pohuashanensis* embryos, the germination percentage was significantly higher in the embryos under ETH treatment compared with those under CK treatment; here, the activities of NO, NR, and NOS were also higher in the CK treatment compared with the saline-alkaline environment. This finding implies that RNS positively regulated the exogenous ethylene-mediated promotion of embryo germination in *S. pohuashanensis*.

## 4. Materials and Methods

### 4.1. Experimental Materials

Mature berries of adult *S. pohuashanensis* plants were collected from the Maoershan Forest Research Station of Northeast Forestry University, Heilongjiang, China (127°30′–127°34′ E, 45°21′–45°25′ N) in early October 2020. Then, mature and infection-free seeds (9–10% water content) were collected from these berries, placed in sealed plastic bags, and stored in the refrigerator at 4 °C for further analysis.

### 4.2. Experimental Methods

#### 4.2.1. Seed Pre-Treatment

Dormant seeds of uniform texture with an average weight of 2.04 mg, an average moisture content of 5.79%, and an average viability of 88.89% were selected, soaked in distilled water at 20 ± 5 °C for 48 h and 0.2% (*v/v*) NaClO solution for 10–15 min, and finally rinsed with water. The coat of these disinfected seeds was stripped on ice (to avoid enzyme inactivation), and the peeled embryos were used for the subsequent experiments [14]. The embryo obtained after pre-treatment is shown in Figure 7.

#### 4.2.2. Ethephon Treatment

Initially, the embryos of *S. pohuashanensis* were treated with different concentrations of ethephon (30, 45, and 60 mg/L; Shanghai Huayuan Chemical Industry, Shanghai, China). Approximately 3 mL of the prepared ETH solution was added to wet the filter paper in a petri dish. The concentration that resulted in the highest germination percentage (45 mg/L) was selected for further analysis.

#### 4.2.3. Seed Germination under Salt Stress

NaHCO_3_ (Tianjin third factory) was chosen to simulate a saline-alkaline stress environment for *S. pohuashanensis* in this study. The test concentration was determined based on approximately 0.15% soluble salt (Na^+^) content present in the saline-alkaline soil surface after the end of the test (concentration of 18 mmol/L; EC = 0; pH = 9.7). NaHCO_3_ and ETH to treat seeds were prepared at a mass ratio of 30:1.

#### 4.2.4. Germination Experiments

The embryos of *S. pohuashanensis* were treated with distilled water, ethephon, NaHCO_3_, or a mixture of NaHCO_3_ and ethephon. Two layers of filter paper were placed in Petri dishes with a diameter of 9 cm [40]. Three microliters of one of the above solutions were added to moisten the filter paper, and the embryos were placed on the filter paper at the rate of 30 embryos per Petri dish. These seeds were incubated at 25 °C under 60 μmol·m^−2^·s^−1^ light (16 h per day) to allow germination. Sterile water was sprayed on the culture dish every 24 h till the end of the germination test to maintain humidity. At least three replicates were maintained per treatment. The germination indexes were calculated using the following equations:Mean germination speed (days) = ∑ (D ∗ n)/∑n
where D represents the number of days from embryo placement, and n indicates the number of germinating embryos on each corresponding day.
Germination percentage (%) = (Number of embryos germinated/Total number of embryos) ∗ 100%
Germination index = ∑Gt/Dt
where Gt indicates the number of germinations on the day, and Dt indicates the number of germination days
Germination potential (%) = (n/N) ∗ 100%
where n is the number of germinated embryos at the peak of daily germination during germination, and N is the total number of embryos.

#### 4.2.5. Endogenous Hormone Assay

The levels of 1-aminocyclopropane-1-carboxylic acid (ACC), gibberellin (GA), and abscisic acid (ABA) in embryos or seedlings were analyzed using specific kits at the Shanghai Enzyme Link Biotechnology Co., Ltd. (Shanghai, China). Standards of known concentration of the substance and samples of unknown concentration were added to the microtiter plate and incubated with antibody and the biotin-labeled antibody. After washing, the affinity-labeled horseradish peroxidase (HRP) was added, and the unbound enzyme conjugate was removed by warming and washing. Then, substrates A and B, which act simultaneously with the enzyme conjugate, were added. The color is proportional to the concentration of the substance to be measured in the sample.

The concentration of each substance was calculated from ELISA results based on the logit curve. The natural logarithm of the hormone’s concentration (ng/mL) was plotted as the horizontal coordinate and the logit value of the color development value for each concentration as the vertical coordinate. Then, calculated as follows:Logit (B/B_0_) = ln (B/B_0_)/(1 − (B/B_0_)) = ln B/(B_0_ − B)
where B_0_ is the color development value for the 0 ng/mL well, and B is for the other concentrations. The concentration of hormone (ng/mL) in the sample was determined based on the logit value of its color development value.

#### 4.2.6. ROS Accumulation Assay

The levels of superoxide anion (O^−^_2_) and hydrogen peroxide (H_2_O_2_) in embryos or seedlings were determined according to the method by Yang L [23]. The content of malondialdehyde (MDA) was determined using thiobarbituric acid (TBA).

#### 4.2.7. Antioxidant Enzyme Activity Assay

The activities of SOD and POD and the concentration of CAT in embryos or seedlings were determined using specific detection kits (Suzhou Comin Biotechnology Co. Ltd., Suzhou, China), following the manufacturer’s instructions. The soluble protein content of the seeds was determined by the Coomassie brilliant blue method [14].

#### 4.2.8. Reactive Nitrogen Species Accumulation Assay

The NO and nitrate reductase (NR) levels in embryos or seedlings were measured using specific kits from Suzhou Comin Biotechnology Co., Ltd., while activity nitric oxide synthase (NOS) was measured using a kit from Nanjing Cheng Jian Institute of Biotechnology.

#### 4.2.9. Data Analysis

In this study, Excel 2003 software was used for data sorting, SigmaPlot v. 12.5 for drawing, and SPSS v. 19.0 for the variance analysis, Duncan’s multiple range test, and correlation analysis. Analysis of variance was used to compare the treatment means, and Duncan’s multiple range test was used to test the differences between the treatment means at the significance level of *p* = 0.05.

## 5. Conclusions

Our study demonstrated that ETH promotes the germination of *S. pohuashanensis* embryos. The saline environment under NaHCO_3_ treatment inhibited the germination of *S. pohuashanensis* embryos, while ETH mitigated the inhibitory effect of NaHCO_3_. The alleviating effect of ETH was mainly driven by regulating hormones, clearing ROS, and improving NO in embryos. These results provide a scientific basis for establishing exogenous ETH treatment to promote the germination of *S. pohuashanensis* embryos and the cultivation of *S. pohuashanensis* in saline-alkaline land.

## Figures and Tables

**Figure 1 ijms-24-04244-f001:**
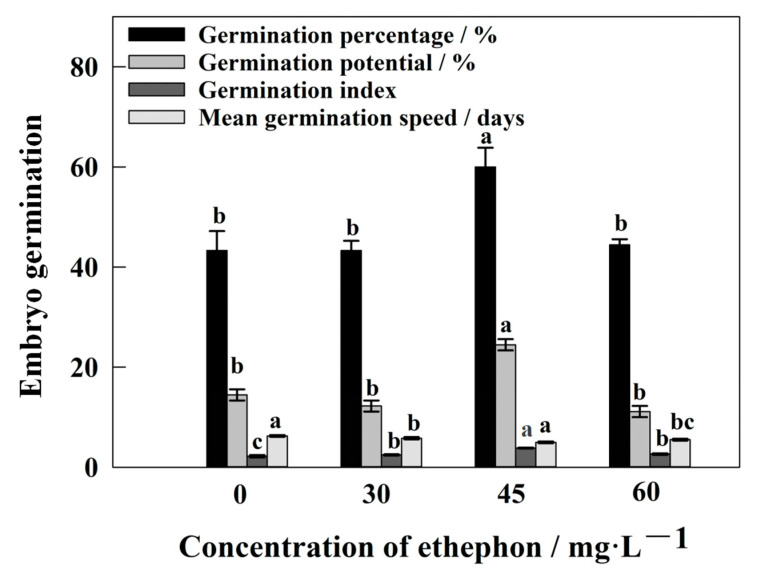
Effect of different concentrations of ethephon on embryo germination of *S. pohuashanensis*; data include mean values ± SE (n = 3); letters indicate significant differences at *p* = 0.05.

**Figure 2 ijms-24-04244-f002:**
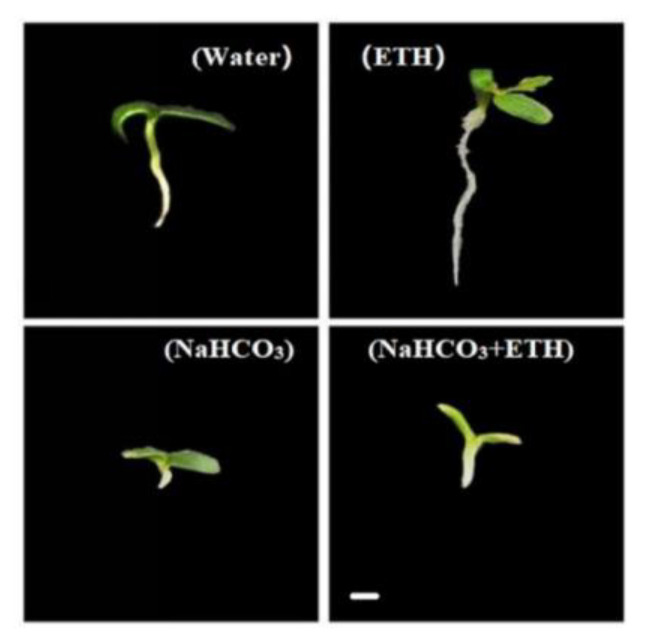
Growth of seedlings on day 8 under various treatments. (Bar = 1.0 mm).

**Figure 3 ijms-24-04244-f003:**
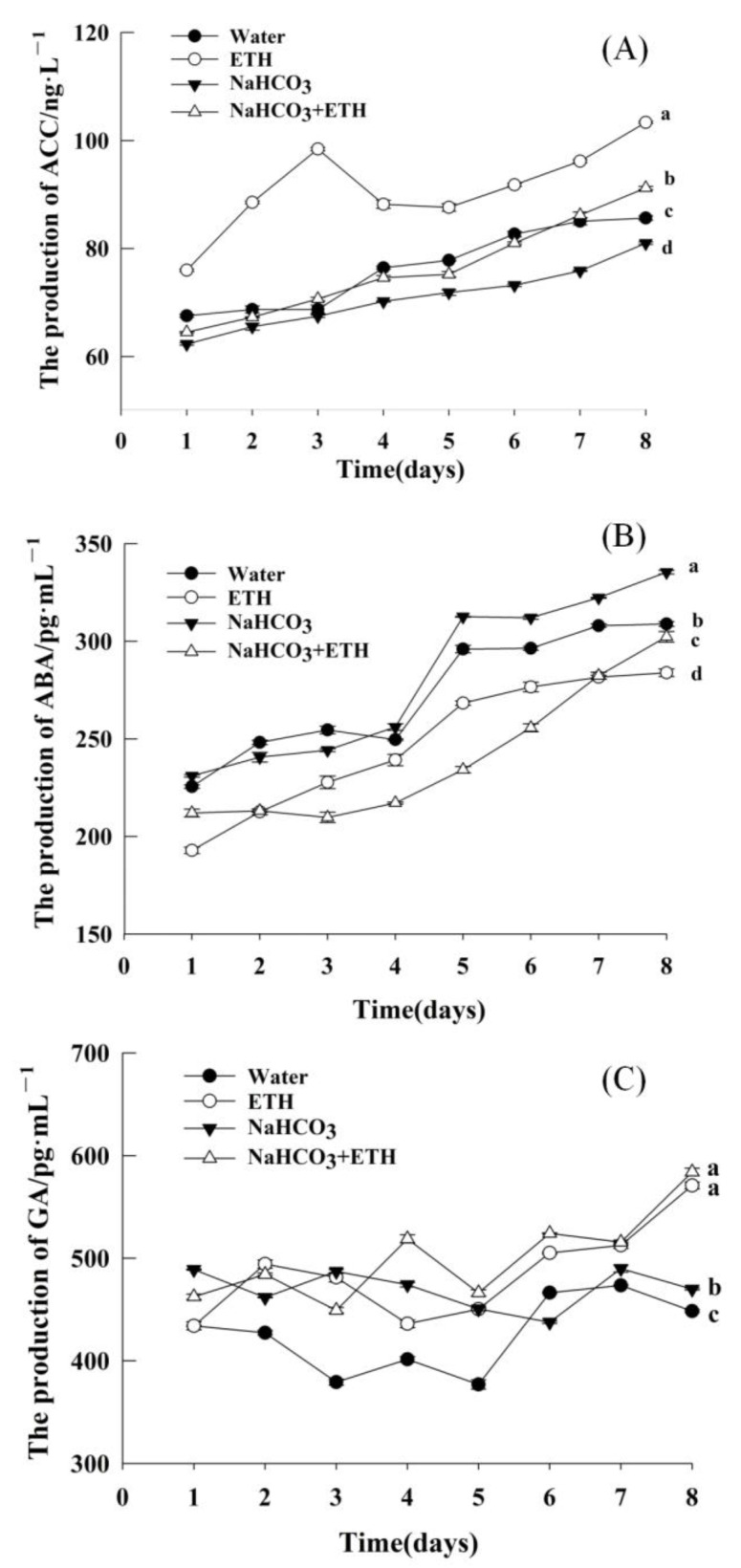
Endogenous hormone content of *S. pohuashanensis* embryos during germination. (**A**) 1-aminocyclopropane-1-carboxylic acid (ACC); (**B**) abscisic acid (ABA); (**C**) gibberellin (GA). Data presented are mean values ± SE (n = 3); different lowercase letters indicate significant differences at *p* = 0.05 on day 8.

**Figure 4 ijms-24-04244-f004:**
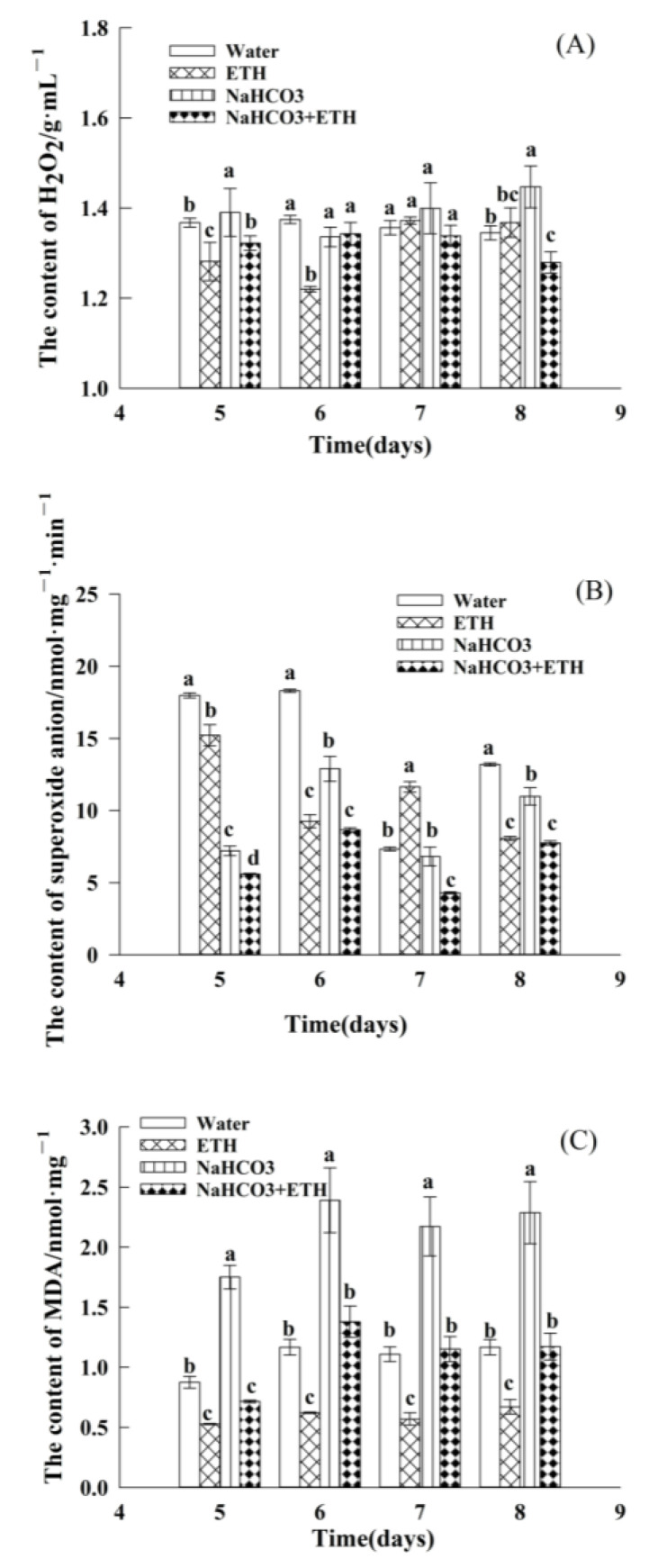
ROS-related indicators in the seedlings of *S. pohuashanensis* from day 5 to 8 of germination. (**A**) Hydrogen peroxide (H_2_O_2_); (**B**) superoxide anion; (**C**) malondialdehyde (MDA). Data presented are mean values ± SE (n = 3); different lowercase letters indicate significant differences at *p* = 0.05 on day 8.

**Figure 5 ijms-24-04244-f005:**
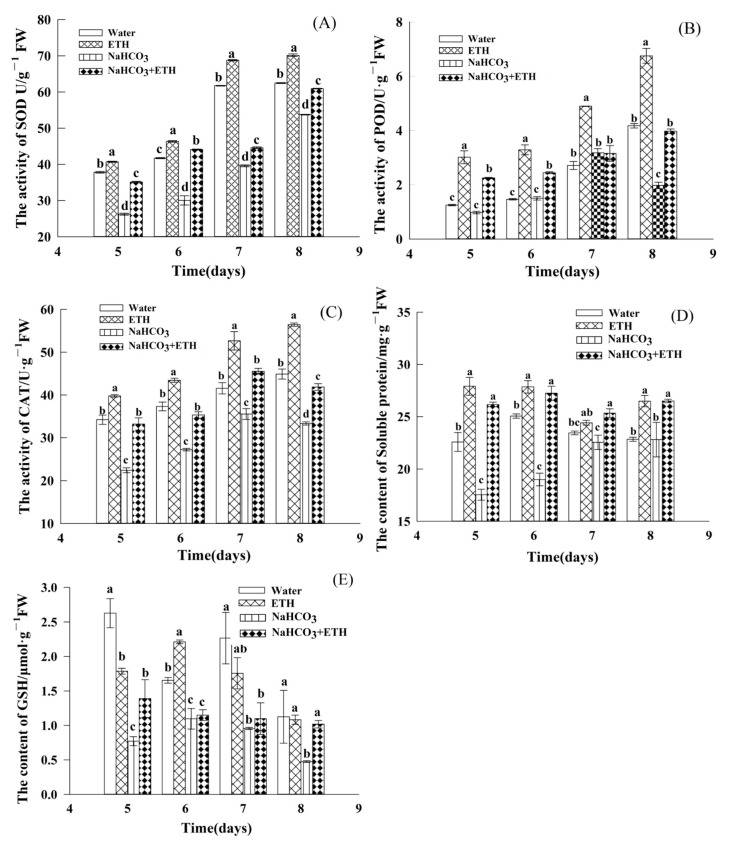
Levels of antioxidant enzymes and related indicators in the seedlings of *S. pohuashanensis* from day 5 to 8 of germination. (**A**) Superoxide dismutase (SOD); (**B**) peroxidase (POD); (**C**) catalase (CAT); (**D**) soluble protein; (**E**) glutathione (GSH). Data presented are mean values ± SE (n = 3); different lowercase letters indicate significant differences at *p* = 0.05 for day 8.

**Figure 6 ijms-24-04244-f006:**
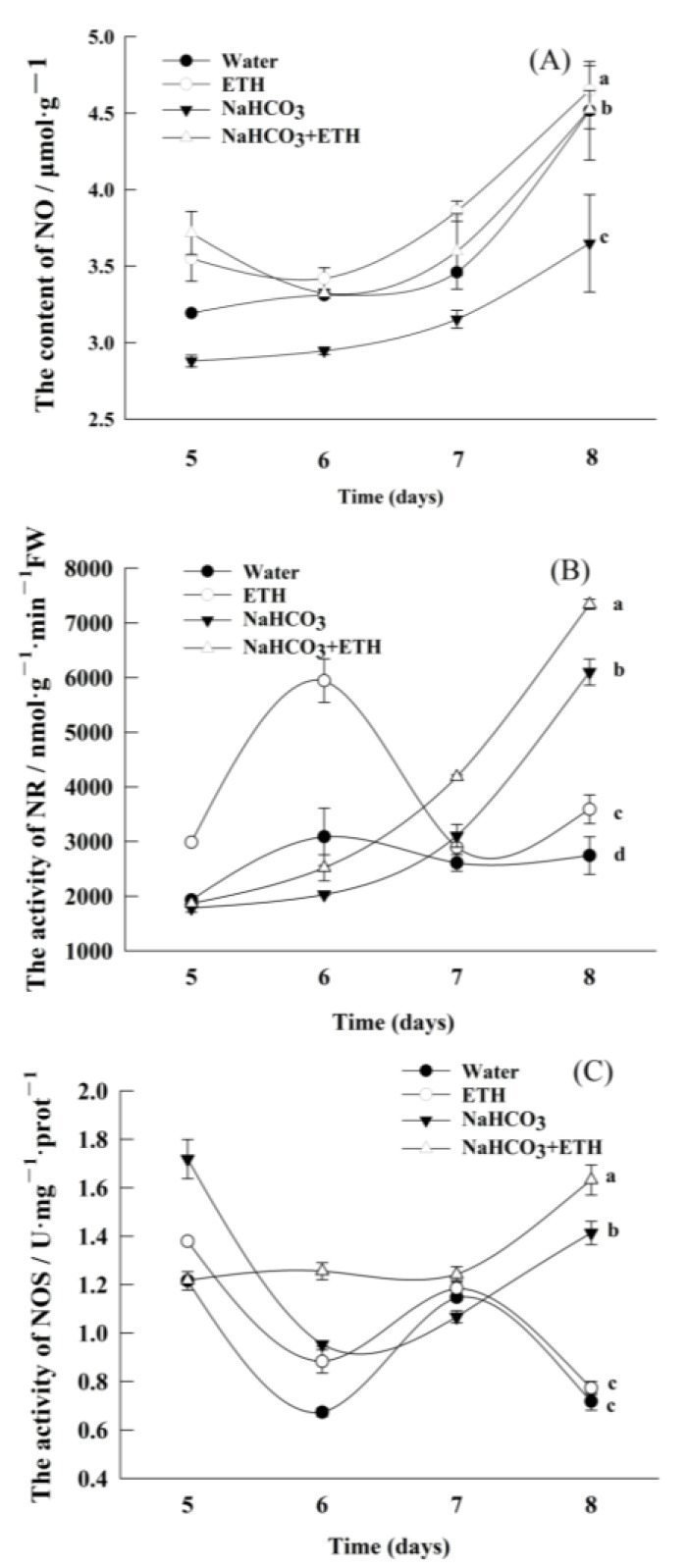
Levels of active nitrogen-related indicators in the seedlings of *S. pohuashanensis* from day 5 to 8 of embryo germination. (**A**) Nitric oxide (NO); (**B**) nitrate reductase (NR); (**C**) nitric oxide synthase (NOS). Data presented are mean values ± SE (n = 3); different lowercase letters indicate significant differences at *p* = 0.05 on day 8.

**Figure 7 ijms-24-04244-f007:**
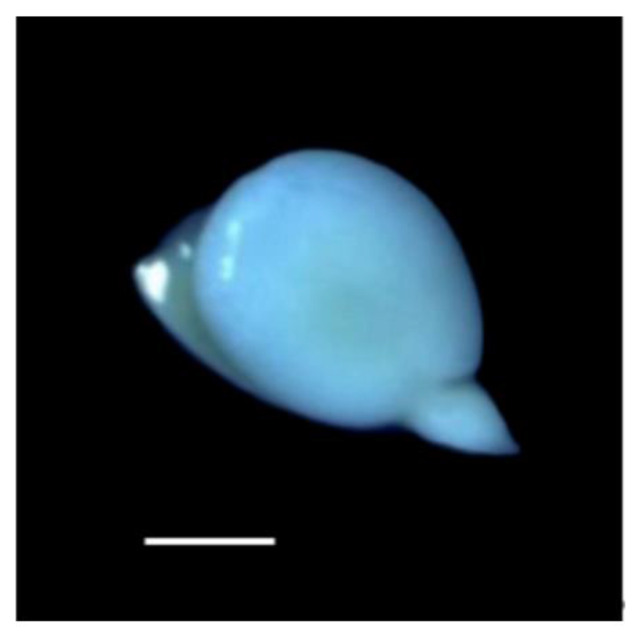
Embryo of *S. pohuashanensis* with seed coat removed (Bar = 1.0 mm).

**Table 1 ijms-24-04244-t001:** Effect of saline-alkali stress and ethephon treatment on the germination indexes of *S. pohuashanensis* embryos. Here, 45 mg/mL ETH was used. Data presented are mean values of three independent experiments; different lowercase letters indicate significant differences at the *p* = 0.05 level.

Index	Water	ETH	NaHCO_3_	NaHCO_3_ + ETH
Germination percentage (%)	43.33 ± 3.95 ^b^	60.00 ± 3.85 ^a^	6.67 ± 1.92 ^d^	18.89 ± 4.01 ^c^
Mean germination speed	6.25 ± 0.16 ^b^	4.98 ± 0.18 ^c^	7.17 ± 0.44 ^a^	6.68 ± 0.09 ^ab^
Germination index	2.19 ± 0.24 ^b^	3.84 ± 0.10 ^a^	0.29 ± 0.09 ^d^	0.87 ± 0.18 ^c^
Germination potential (%)	14.44 ± 1.11 ^b^	24.44 ± 1.11 ^a^	0.29 ± 0.10 ^c^	6.67 ± 1.92 ^c^

For each parameter, means followed by different lowercase letters are significantly different according to Duncan’s *t*-test (*p* < 0.05) (mean ± SD; n = 3).

## Data Availability

Not applicable.

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
