# Peer review of "Exogenous Ethylene Alleviates the Inhibition of Sorbus pohuashanensis Embryo Germination in a Saline-Alkali Environment (NaHCO3)"

_ijms, 2023, doi:10.3390/ijms24044244_

Round 1

Reviewer 1 Report

Main comments:

The research undertaken by the authors of the manuscript may be of interest because of the important issue of improving germination, using ethephon as an ethylene donor, especially under environmental stresses. The physiological role of ethylene in seed germination and its effect on metabolic processes is not well understood, so I find the findings presented in the manuscript under review to be entirely useful. The paper is reasonably well written however, I suggest corrections and additions. My comments mainly concern the research methodology and clarity of issues.

Detailed comments and suggestion:

Abstract:

Line 14-17 The abstract seems to be long and some sentences are repeated. I suggest to leave this information: “and severely limits … mechanism”

Line 18-19: Also leave this information: “which … species (RNS)”

Line 20: should be added: ethephon as the exogenous ethylene donor

Line 26-27: it would be necessary to specify how exogenous ethylene affects other germination parameters (mean germination speed, germination index, germination potential)

Keywords: I propose to omit the words appearing in the title and add seed germination parameters and phytohormones.

Introduction

Line 64: ethephon as the exogenous ethylene donor

Results:

Line 97: The effect of the addition of ethephon as the exogenous ethylene donor

Line 100: different concentrations of ethephon. Figure 1A is repeated with the results in figure 1D. Figure 1A would have to be omitted.  All germination parameters should also be described in detail (mean germination speed, germination index, germination potential)

Line 102: with ethephon

Line 114-117: Figure 1: leave figure 1A.

Discussion:

Line 241: ETH

Materials and Methods:

Those parameters were calculated using the following equations:

Line 312: Should be: most suitable germination concentration being selected (45 mg/L).

Line 327: How were the different germination parameters determined (mean germination speed, germination index, germination potential)? calculations or equations needed.

Author Response

Dear Reviewer,

We are very grateful to you for taking the time to read and modify our article again. We find that your comments play a very important role in improving the quality of our papers. We have carefully revised the paper in light of your comments, and please find our response to the comments made below. We marked the modified part of the manuscript in red.

Thank you for considering our revised manuscript!

Point 1: Line 14-17: The abstract seems to be long and some sentences are repeated. I suggest to leave this information: “and severely limits … mechanism”

Response 1: Thank you very much for your suggestion. We read the abstract carefully and accepted your suggestions. For details please see the abstract.

Point 2: Line 18-19: Also leave this information: “which … species (RNS)”

Response 2: Thank you very much for your suggestion. We read the abstract carefully and revised sentences of abstract. For details please see the abstract.

Point 3: Line 20: should be added: ethephon as the exogenous ethylene donor

Response 3: Thank you very much for your recognition. We read the abstract carefully and accepted your suggestions. For details please see the abstract.

Point 4: Line 26-27: it would be necessary to specify how exogenous ethylene affects other germination parameters (mean germination speed, germination index, germination potential)

Response 4: Thank you very much for your suggestion. We read the abstract carefully and added other germination parameters. For details please see the abstract.

Point 5: Keywords: I propose to omit the words appearing in the title and add seed germination parameters and phytohormones.

Response 5: Thank you very much for your suggestion. We read the keywords carefully and accepted your suggestions. For details please see the keywords.

Point 6: Line 64: ethephon as the exogenous ethylene donor

Response 6: Thank you very much for your suggestion. We read the manuscript carefully and accepted your suggestions. For details please see the text (line 61).

Point 7: Line 97: The effect of the addition of ethephon as the exogenous ethylene donor

Response 7: Thank you very much for your suggestion. We read the manuscript carefully and accepted your suggestions. For details please see the text (line 90).

Point 8: Line 100: different concentrations of ethephon.

Response 8: Thank you very much for your suggestion. We read the manuscript carefully and accepted your suggestions. For details please see the text (line 89).

Point 9: Figure 1A is repeated with the results in figure 1D. Figure 1A would have to be omitted.  All germination parameters should also be described in detail (mean germination speed, germination index, germination potential)

Response 9: Thank you very much for your suggestion. We read the manuscript carefully and accepted your suggestions. Figure 1A has been deleted and Figure 1B has been added according to your suggestion. For details please see the figure 1.

Point 10: Line 102: with ethephon

Response 10: Thank you very much for your suggestion. We read the manuscript carefully and accepted your suggestions. For details please see the text.

Point 11: Line 114-117: Figure 1: leave figure 1A.

Response 11: Thank you very much for your suggestion. We read the manuscript carefully and accepted your suggestions. For details please see the figure 1.

Point 12: Line 241: ETH

Response 12: Thank you very much for your suggestion. We read the manuscript carefully and accepted your suggestions. For details please see the text.

Point 13: Materials and Methods:

Those parameters were calculated using the following equations:

Response 13: Thank you very much for your suggestion. We read the manuscript carefully and accepted your suggestions. For details please see the Materials and Methods.

Point 14: Line 312: Should be: most suitable germination concentration being selected (45 mg/L).

Response 14: Thank you very much for your suggestion. We read the manuscript carefully and accepted your suggestions. For details please see the Materials and Methods Line 316.

Point 15: Line 327: How were the different germination parameters determined (mean germination speed, germination index, germination potential)? calculations or equations needed.

Response 15: Thank you very much for your suggestion. We read the manuscript carefully and accepted your suggestions. For details please see the Materials and Methods Lines 334-344.

Reviewer 2 Report

The manuscript presents the evaluation of Ethylene as an alternative to reduce the effect of saline-alkali stress in the early stages of germination. The results are important for the contribution to the knowledge of the enzymatic mechanisms of response to salt stress and of great application for the management and propagation of Sorbus pohuashanensis.

Further discussion of the optimal effect found in ETH is suggested, since with 45 mg/L the performance of embryos and seeds improves, but with 60 mg/L it is reduced to the initial control values, as shown in Figure 1.

Additionally, the following changes are suggested:

Page 1, line 14, remove "under saline alkaline soil conditions".

Page 1, line 15, remove "but"

Page 1, line 16, change "urgent" to "necessary".

Page 4, line 123, indicate in the header of the table that the ETH concentration is 45 mg/L.

Pages 3-4, lines 114-124, the information in Figure 1 and Table 1 is repeated, consider leaving the table or figure.

Page 9, line 251, change "eth" to "ETH".

Page 11, line 311, indicate how the ETH concentrations evaluated were defined.

Page 11, lines 330-331, Briefly describe the Shanghai enzyme-linked kit for high-performance liquid chromatography technique, to understand how an enzymatic method is coupled with a chromatographic system to arrive at the pg sensitivity for the case of GA (Figure 1).

Author Response

Dear Reviewer,

We are very grateful to you for taking the time to read and modify our article again. We find that your comments play a very important role in improving the quality of our papers. We have carefully revised the paper in light of your comments, and please find our response to the comments made below. We marked the modified part of the manuscript in red.

Thank you for considering our revised manuscript!

Point 1: Further discussion of the optimal effect found in ETH is suggested, since with 45 mg/L the performance of embryos and seeds improves, but with 60 mg/L it is reduced to the initial control values, as shown in Figure 1.

Response 1: We carefully studied your suggestion and agreed that what you said is very reasonable. A discussion on the effect of ethephon concentration on germination has been added. For details please see the text (Line 224-228).

Point 2: Page 1, line 14, remove "under saline alkaline soil conditions".

Response 2: We carefully studied your suggestion and agreed that what you said is very reasonable. For details please see the Abstract.

Point 3: Page 1, line 15, remove "but"

Response 3: We read the manuscript carefully and accepted your suggestions. For details please see the Abstract.

Point 4: Page 1, line 16, change "urgent" to "necessary".

Response 4: We read the manuscript carefully and accepted your suggestions. For details please see the Abstract.

Point 5: Page 4, line 123, indicate in the header of the table that the ETH concentration is 45 mg/L.

Response 5: We read the manuscript carefully and accepted your suggestions. For details please see the table 1.

Point 6: Pages 3-4, lines 114-124, the information in Figure 1 and Table 1 is repeated, consider leaving the table or figure.

Response 6: Thank you very much for your suggestion. We read the manuscript carefully, and accepted your suggestions. Repeated Figure have been deleted, for details please see the figure 1.

Point 7: Page 9, line 251, change "eth" to "ETH".

Response 7: Thank you very much for your suggestion. We read the manuscript carefully and accepted your suggestions. For details please see the text (Line242).

Point 8: Page 11, line 311, indicate how the ETH concentrations evaluated were defined.

Response 8: Thank you very much for your suggestion. We read the manuscript carefully and accepted your suggestions. For details please see the text (Line 314-315).

Point 9: Page 11, lines 330-331, Briefly describe the Shanghai enzyme-linked kit for high-performance liquid chromatography technique, to understand how an enzymatic method is coupled with a chromatographic system to arrive at the pg sensitivity for the case of GA.

Response 9: Thank you very much for your suggestion. We read the manuscript carefully . Your suggestions have been accepted and changes have been made. Here is my mistake, should be enzyme method, has been corrected, for details please see the Materials and Methods (4.2.5. Endogenous hormone assay).

Reviewer 3 Report

The paper lacks quality in its writing-  needs professional editing   made some suggestions but the whole paper requires work on wording

the methods descriptions are inadequate for reproduction

the measurement of ethylene  in the treated systems is needed  (more work)

other comments for additions are found in sticky notes

Author Response

Dear Reviewer,

We are very grateful to you for taking the time to read and modify our article again. We find that your comments play a very important role in improving the quality of our papers. We have carefully revised the paper in light of your comments, and please find our response to the comments made below. We marked the modified part of the manuscript in red.

Thank you for considering our revised manuscript!

Point 1: The paper lacks quality in its writing- needs professional editing made some suggestions but the whole paper requires work on wording

Response 1: Thank you very much for your suggestion. We revised the English of the article. The authors would like to thank Top Edit (www.topeditsci.com) for its linguistic assistance during the preparation of this manuscript.

Point 2: the methods descriptions are inadequate for reproduction

Response 2: Thank you very much for your suggestion. We read the manuscript carefully and accepted your suggestions. The test method is supplemented in the article, for details please see the text (4.2. Experimental method, line 295-301).

Point 3: measurement of ethylene in the treated systems is needed (more work)

Response 3: Thank you very much for your suggestion. We did not directly assay the ethylene content, we measured the endogenous ACC content of the different treated samples in this paper. ACC is a synthetic precursor of ethylene and we used ACC content to indicate ethylene content in this paper.

Point 4: first sentence must be reworked too long

Response 4: Thank you very much for your suggestion. We read the manuscript carefully and accepted your suggestions. For details please see the Abstract.

Point 5: the abstract requires extensive professional editing removal of repetitions etc

Response 5: Thank you very much for your suggestion. We read the manuscript carefully and accepted your suggestions. For details please see the Abstract.

Point 6: limiting the activity of

Response 6: Thank you very much for your suggestion. We read the manuscript carefully and accepted your suggestions. For details please see the text (line 41).

Point 7: reduces

Response 7: Thank you very much for your suggestion. We read the manuscript carefully and accepted your suggestions. For details please see the text (line 46).

Point 8: where is malate accumulated

Response 8: Thank you very much for your suggestion. We read the manuscript carefully. Original content : Under sodium bicarbonate (NaHCO3) stress, wheat was found to accumulate malic acid by up-regulating malate dehydrogenase (MDH) and down-regulating malate enzyme and enzymes related to the tricarboxylic acid cycle (TCA cycle) to cope with alkaline stress [6]. Malic acid is synthesized and metabolized in the cytoplasm of plant fruit pulp cells and accumulated in large quantities in the vesicles. However, the mechanism of malic acid accumulation and metabolic regulation in seeds is not clear and requires experimental studies to elucidate.

Point 9: is MDA or what MDA accumulation stands for the reason for toxicity?

Response 9: Thank you very much for your question. Malondialdehyde (MDA) is one of the common indicators of oxidative stress and reflects the degree of membrane lipid peroxidation in plants. In living organisms, free radicals act on lipid peroxidation, and the end product of oxidation is malondialdehyde, which causes cross-linking and polymerization of proteins, nucleic acids, and other life macromolecules, and is cytotoxic. The extent of membrane lipid peroxidation can be indirectly measured by MDA. We changed this sentence (line 156).

Point10: do not understand sentence

Response 10: Thank you very much for your question. We modified the language expression.

Point 11: other hormones? 

Response 11: Thank you very much for your suggestion. We focus on the synthesis and metabolism of ethylene in this paper.

Point 12: could be better worded

Response 12: Thank you very much for your question. We modified the language expression.

Point 13: unclear what is meant

Response 13: Thank you very much for your question. We modified the language expression.

Point 14: must show error think 2 dp is not needed:These results indicated that ethephon reduced the inhibitory effect of saline-alkali stress on the germination of S. pohuashanensis embryos (Fig. 1C, D).

Response 14: Thank you very much for your suggestion. We read the manuscript carefully and accepted your suggestions. For details please see the text (Fig 1).

Point 15: where in the embryo are the cells producing ACC?could their be a role of limiting the products of ACC deamination eg alphaketo butyrate. what happens to amino acid levels?

Response 15: Thank you very much for your question. At present, it is not clear where the cells that produce ACC in plant embryos are. NaHCO3 stress can limit the products of ACC deamination, the effect on amino acid levels is unclear. These questions are very interesting and worthy of our consideration and future research.

Point 16:   Fig2 do not understand your stats     are all points significantly different…. better to show as bar graph with each bar having stat analysis

Response 16: Thank you very much for your question. Because our data is the result of continuous observation, we use a line chart instead of a column chart. The letters in Figure 3 represent the statistical analysis results on the 8th day. We have supplemented the explanation in the caption. For details please see the Fig3 (line 137).

Point 17: how is GA mass/ml assessed do not understand the ml ' must be related back to mass of embryo 

Response 17: Thank you very much for your question. Regarding the measurement of hormones, we used an ELISA and the relevant principles and calculations have been added to the text (lines 347-363).

Point 18: should really look at different isozymes rather than total

Response 18: Thank you very much for your question. We did not observe the different isozyme changes, but only the total enzyme activity, which we will add in a future study.

Point 19: but how do you normalize the data must explain

Response 19: Thank you very much for your question. We proofread the data and unit.

Point 20: why is NR looked at needs explanation

Response 20: Thank you very much for your suggestion. We read the manuscript carefully and accepted your suggestions. For details please see the text (lines 203-204). NR is the key to the conversion of nitrate nitrogen to ammonia nitrogen in plants and has an impact on plant growth and development. NR is one of the sources of nitric oxide signaling molecules in plants, and the enhanced NR activity suggests that ETH promotion of Sorbus embryo germination may be closely related to nitric oxide signaling because NR is a source of intracellular nitric oxide signaling molecule production.

Point 21: how do you know it is salinity rather than alkalinity

Response 21: Thank you very much for your question. Here is my poor consideration, has been modified in the article (line 224-228).

Point 22: Is ACC really a hormone would not define it as such

Response 22: Thank you very much for your question. The ethylene precursor ACC may be a novel hormone signaling molecule. For details please see the text (lines 235-239).

Point 23: confused writing: In this study, ACC was released continuously with germination time under ETH treatment (with the highest content of ACC), while the lowest amount of ACC was released from NaHCO3 treated S. pohuashanensis embryos. Meanwhile, ETH increased the release of ACC from embryos under NaHCO3 treatment.

Response 23: Thank you very much for your suggestion. We read the manuscript carefully and accepted your suggestions. Revisions have been made in the text (lines lines 235-239).

Point 24: eth? 

Response 24: Thank you very much for your suggestion. We read the manuscript carefully and accepted your suggestions. For details please see the text (line 242).

Point 25: ? ??  wording ? :  The seeds were prepared by the water separation method, and full, pure, mature, and safe water content (9%-10%) seeds were placed in plastic bags and stored in the refrigerator at 4ËšC for further analyses.

Response 25: Thank you very much for your suggestion. We read the manuscript carefully and accepted your suggestions. For details please see the Materials and Methods (lines 304-309).

Point 26: what is EC and pH no Ca or Mg?

Response 26: Thank you very much for your question. EC= 0; pH =9.7 (line 340 ) , the most important effect on plant growth and development in saline lands is Na, so Ca or Mg are not considered here.

Point 27: inadequate descriptions: Endogenous hormone assay

Response 27: Thank you very much for your suggestion. We read the manuscript carefully and accepted your suggestions. For details please see the Materials and Methods(lines 347-363).

Point 28: improving what does this mean up or down: The alleviating effect of ETH was mainly manifested in regulating hormones, clearing the accumulation of ROS in the embryos, and im-proving the level of RNS in embryos.

Response 28: Thank you very much for your question. This implies that RNS plays an important positive regulatory role in the exogenous ethylene promotion of embryo germination in rowan. We have added notes at the appropriate places in the text (lines 292-294).

Point 29:  you could add more information about these saline alkaline soils

need descriptions of soil properties 

Response 29:  Thank you very much for your question. We will pay more attention to the characteristics of saline soils in our future studies.
